# Productivity Change and Decomposition in Taiwan Bakery Enterprise—Evidence from 85 °C Company

**Chieh-Wen Chang [1,2], Kun-Shan Wu [3,*] and Bao-Guang Chang [4]**

[1]  Department of Management Sciences, Tamkang University, Taipei 25137, Taiwan; 85cryan@gmail.com
[2]  Gourmet Master Co., Ltd, Taichung 408, Taiwan
[3]  Department of Business Administration, Tamkang University, Taipei 25137, Taiwan
[4]  Department of Accounting, Tamkang University, Taipei 25137, Taiwan; baog@mail.tku.edu.tw
[*]  Correspondence: kunshan@mail.tku.edu.tw; Tel.: +886-2-2621-5656 (ext. 3374)

**Abstract:** In recent years, the bakery market has grown rapidly. Alongside its growth and fast change, it is very important to comprehend the productivity change of the bakery industry. Nowadays, effective management is more and more important to ensure the sustainable development of enterprises. Thus, productivity change of 22 self-owned stores of a famous bakery company (85 °C) from 2011 to 2016 was quantitatively analyzed and evaluated by adopting Malmquist index model in this study. Based on the Malmquist index model, the overall mean for total productivity change of 85 °C increased slightly from 2011 to 2016, and the productivity change was easily affected by technical progress. Moreover, the results also show that the north-district self-owned stores (which are located in subtropical climate) have the worst technical progress and total factor productivity change during 2011–2016 period by adopting the non-parametric Kruskal–Wallis and Dunn post-hoc test.

**Keywords:** bakery industry; Malmquist index; productivity change; technical progress; relative efficiency change

## 1. Introduction

Under an intense internationally competitive business environment, it is important to understand the production efficiency of the bakery industry, where production efficient management is becoming increasingly important to ensure the sustainable development of the company. Food industry is not only important for livelihood, but also a key indicator of national development and people's quality of life. The bakery industry is a division of the food industry and consists of companies that produce pastries, cookies, breads, cakes, and other bakery products. Based on the report of Mordor Intelligence [1], the global market size of the bakery products is forecasted to reach USD 543.9 billion by 2024, with a Compound Annual Growth Rate (CAGR) of 2.6% during the forecast period (2019–2024). Hence, the market potential is huge. The Taiwan food industry annual output value was NTD 615.3 billion in 2018, and this number accounts for the total manufacturing output value about 4.17% [2], together with more than 450,000 companies among the food industry, according to the report of Ministry of Economic Affairs (MOEA) in Taiwan.

As for Taiwan, on the basis of the 2018 ITIS report of FIRDI [3] of Taiwan shows that in 2017, there were about 615 bakery companies in Taiwan, and the total revenue was about NTD 70.8 billion, whereas in 2018, there were 672 bakery companies with total revenue of NTD 72.7 billion [1]. The report also revealed that the overall annual output value for Taiwan bakery products has reached NTD 31.3 billion in 2017 and NTD 32.1 billion in 2018, accounting for more than 6% of the total food industry output value.

Searching for a more efficient use of their resources is a common issue in the sustainability goals of any industry [4]. An important key issue for various industries is to identify vulnerabilities. Performing this issue enables companies to get several operation advantages such as knowing how to effectively use their sources, helping estimating their performance in competitive industries, and helping authenticating companies to be used as a reference. In the bakery industry, productivity growth is important because of fierce market competition. Since the baking business has taken an important role in the Taiwanese economy, and the profitability of Taiwanese bakery companies are significantly affected by its productivity growth, it is necessary to maintain their competitiveness and sustainability.

Although some studies have conducted extensive discussion on the efficiency of Taiwan's tourism industry [5–8], we find that the study on the productivity of Taiwan's bakery industry is rare. Recently, Chang, Wu, Chang, and Lou [9] applied Data Envelopment Analysis (DEA) to evaluate the performance of a famous bakery enterprise (85 °C) on Taiwan Stock Exchange (TSE) during 2011 to 2016, and used input-output structure measurement of technical efficiency and scale efficiency to recognize the main causes of efficiency loss. However, Chang et al. [9] made horizontal comparison and analysis on the relative efficiency of 85 °C company in specific years, while this study will continuously supplement the in-depth research on the dynamic change of productivity.

Therefore, this article will be on the basis of the existing literature, further enrich the research content, and expand the study of 85 °C company productivity growth perspective. The main purpose of the article is to evaluate the performance assessment of the famous baking company, 85 °C, listed on TSE during 2011 to 2016, and to estimate the productivity growth of this company with DEA-Malmquist Index. In addition, this study constructed a management decision matrix based on relative efficiency change and total factor productivity change similar to BCG matrix to facilitate the decision-making of managers. The matrix reflects the sustainable development potential of each self-owned store and also shows the spot competitiveness.

The contribution of this paper is that the productivity change has been the first time analyzed by applying DEA-Malmquist model in baking industry as using DEA to measure bakery industry productivity change have not been found. Moreover, the productivity change analysis also provides some other viewpoint for management level of bakery or food companies as through the Malmquist productivity index helps managers to identify strategic key success point for retail stores restructured or business model changed.

## 2. Literature Review on Malmquist Total Factor Productivity Index

Throughout the literature, there are few studies on the productivity change of baking industry or companies, while there are many studies on the productivity change measurement of food producing industry or companies. Meanwhile, Malmquist total factor productivity index and related methods have been used by many studies to measure the productivity change of the food manufacturing in different settings in recent decades.

In the context of Australian dairy processing industry, Doucouliagos and Hone [10] used the SFA to derive Malmquist index in order to estimate and explore trends in the economic performance. They found that the productivity of dairy processing industry has moderate growth, mainly driven by technical progress. The European dairy farms, which was analyzed by Madau, Furesi, and Pulina [11], were discovered the total factor productivity (TFP) change and a decline in productivity in the European milk industry. Tatli and Bayrak [12] explored the TFP change of 22 listed companies in Istanbul stock exchange (BIST) between 2011 to 2015 by applying the Malmquist index to data with input-oriented models. The main results show that all firms seem to decline only in technical progress, while relative efficiency, pure technical efficiency, scale efficiency, and total factor productivity change are opposite. Furthermore, Pongpanich, Peng, and Wongchai [13] used the Slacks-based Measure context-dependent Data Envelopment Analysis (SBM context-dependent DEA) to evaluate the efficiency of the Thailand agriculture and food industry during 2011–2014. They also used the Malmquist index to illustrate the changes of total productivity in the industry. The results of this article enable the enterprises inside

and outside the stock exchange of agriculture and food industry to achieve the performance level and benchmark, so as to improve the operating level of enterprises.

Furthermore, the food manufacturing industry in China has been targeted by some scholars and focus on investigating and studying its productivity change. Yang, Zhang, and Wang [14] applied the Malmquist index to calculate the TFP change of the food processing industry in Jilin Province, China from 2006–2010. The results indicate that productivity changes of Jilin food processing industry had increased rapidly from 2006 to 2010, and the average annual growth rate of TFP was 17.5%. The technical progress of food processing industry in Jilin province grows slowly, and the technical progress become the main factor to promote the growth of TFP. Zhang and Wang [15] based on industrial agglomeration, had applied the DEA-Malmquist index to the panel data of 31 provinces in China for the period of 2003–2010, in order to explore the food enterprises scale competitiveness and TFP growth. The main results found that enterprises scale competiveness promoted TFP growth of three food sub-industries, mainly through technical progress. Moreover, Qiang and Fang [16] measured the productivity changes of the food manufacturing industry in China from 2010 to 2014 by using the Malmquist index. The result revealed that the main reason which caused the decrease of TFP is the deterioration of pure technical efficiency.

Meanwhile, as to the Greek food industry, Vlontzos and Theodoridis [17] used the context-depended DEA and Malmquist index to measure the efficiency and the productivity change of dairy firms and found a positive productivity growth. Their results displayed that the average efficiency scores of CRS and VRS DEA models are 0.73 and 0.81, respectively, indicating that there is still room for improvement in the allocation of existing resources. Moreover, productivity change of dairy enterprises increased from 2003 to 2007, with the exception of 2007, as its productivity decreased by 1.9%.

Still in the context of Greece, Giokas, Eriotis, and Dokas [18] used the Malmquist index to examine the liquidity and sales efficiency of food and beverage companies listed on the Athens exchange between 2006 and 2012. An empirical study revealed that the main cause of overall technical inefficiency is pure technical inefficiency, rather than scale inefficiency. Moreover, the results indicate that the productivity of enterprises increased by an average annual rate of 0.5% during the study period.

In India, the Malmquist index had been used by Kumar and Basu [19] to determine the efficiency of the Indian food industry between 1988 to 2005. The results exhibit that the productivity growth of this industry is depressed by negative technical progress and pure technical efficiency loss. Jabir, Singh, and Ekanem [20] have discovered the causes of inefficiency across various sectors by exploring DEA efficiency and Malmquist index between pre- and post-liberalization from 1980–2002. The main findings were that there had been greater changes in efficiency and productivity growth in the high-value sub-sectors of the food processing industry. Likewise, Kaur and Kaur [21] applied the DEA approach and Malmquist TFP change to evaluate, between 1988 to 2011, efficiency changes and performance of various companies in the food processing industry. It is found that the overall average technical progress of the food processing industry is on the decline throughout the study period.

Furthermore, the Malaysia food processing industry was also analyzed by Yodfiatfinda, Mad Nasir et al. [22]. They applied the Malmquist productivity index to discuss the Malaysian food processing of large enterprises (LSEs) productivity growth and efficiency during 2000–2006 periods. The main results showed that the growth of TFP of food processing enterprises in Malaysia is a change of technical progress, but LSEs can also improve TFP growth by promoting production frontier. Additionally in the context of Malaysia, Munshi Naser et al. [23] applied order-m and Malmquist index to explore total factor productivity change in 34 food processing industries using data from 2009–2010. The results demonstrated that almost all industries go through effective technical contribution in their respective production functions, but the technical progress of the organic components of each industry varies greatly.

In addition, Kapelko, Lansink, and Stefanou [24] applied DEA-Malmquist index and its component to calculate the productivity change of the Spanish dairy industry during 1996–2011. The results

suggested that productivity declined on average during the survey period. Moreover, the decomposition of the Malmquist index displays that the technical progress leads to a decrease in productivity.

On the basis of this review, food manufacturing companies or industries in different countries or regions have been discussed or targeted by many researches, with little attention paid to productivity modeling in the baking industry. As many bakery products are essential, the productivity growth of bakery companies or industries is important for the public to know.

## 3. Materials and Methods

### 3.1. DEA-Malmquist Index Model

DEA differs from simple ratio analysis in that it contains multiple inputs and outputs and provides important additional information about where efficiency improvements can be achieved and by how much. DEA is recognized in the literature as a powerful method, which is more suitable for performance measurement activities than traditional econometric methods, such as regression analysis and simple ratio analysis [25–29], and it is also able to determine the operational and managerial efficiencies of companies and industries, etc. [30]. In DEA, each decision-making units (DMU) can freely choose any combination of input and output to maximize its relative efficiency [28].

The Malmquist index is a total factor productivity (TFP) change index obtained from time and cross section data. There are two main approaches to productivity change indicators, either the maximum output for a given input level, or the minimum input for a given output level. These are represented output-oriented and input-oriented measurements, respectively [31].

This study follows Färe, Grosskopf, Norris, and Zhang [32] in their output orientation Malmquist index. This means that the benchmark of productivity change for any data point will be its relative position to the maximum possible output given the level of input. The output-oriented Malmquist index follows the structure of Färe et al. [33], and can be expressed as follows:

$$M_o(x_{t+1}, y_{t+1}, x_t, y_t) = \left[ \frac{D_o^t(x_{t+1}, y_{t+1})}{D_o^t(x_t, y_t)} \times \frac{D_o^{t+1}(x_{t+1}, y_{t+1})}{D_o^t(x_t, y_t)} \right]^{1/2} \tag{1}$$

where $(x_t, y_t)$ denotes the input-output vector at time t, $(x_{t+1}, y_{t+1})$ is the vector at time t + 1; $D_o^t(x_t, y_t)$ represents the output distance function of the input-output vector in t period, and $D_o^t(x_{t+1}, y_{t+1})$ represents the distance from the period t + 1 observation to the period t technology.

$$M_o(x_{t+1}, y_{t+1}, x_t, y_t) = \frac{D_o^{t+1}(x_{t+1}, y_{t+1})}{D_o^t(x_t, y_t)} \left[ \frac{D_o^t(x_{t+1}, y_{t+1})}{D_o^{t+1}(x_{t+1}, y_{t+1})} \times \frac{D_o^t(x_t, y_t)}{D_o^{t+1}(x_t, y_t)} \right]^{1/2} \tag{2}$$

where the ratio of outside the square brackets measures the change in relative efficiency between t and t + 1. The geometric mean of the two ratios inside the square brackets captures the change in technology between the two periods. These may be given as:

$$Efficiency\ change = \frac{D_o^{t+1}(x_{t+1}, y_{t+1})}{D_o^t(x_t, y_t)} \tag{3}$$

$$Technical\ change = \left[ \frac{D_o^t(x_{t+1}, y_{t+1})}{D_o^{t+1}(x_{t+1}, y_{t+1})} \times \frac{D_o^t(x_t, y_t)}{D_o^{t+1}(x_t, y_t)} \right]^{1/2} \tag{4}$$

Thus, the Malmquist total factor productivity change (Tfpch) can be decomposed in to relative efficiency change (Effch) and technical progress (Techch), i.e.,

$$Tfpch = Effch \times Techch \tag{5}$$

Furthermore, relative efficiency change (Effch) can be further decomposed into pure technical efficiency change (Pech) and scale efficiency change (Sech), i.e.,

$$\text{Effch} = \text{Pech} \times \text{Sech} \tag{6}$$

In Equation (5), Techch reflects the technical progress. Pech reflects the technological updating speed of production field, and Sech reveals the influence of input growth on Malmquist index in the Equation (6).

Tfpch > 1 indicates that the TFP change of DMU shows an uptrend from t period to t + 1 period. Tfpch < 1 indicates that TFP change of DMU shows a downward trend from t period to t + 1, while Tfpch = 1 indicates that productivity has no change during this period. The production frontier represents the state of optimum efficiency under prior conditions, so Techch represents the Tfpch changes brought by the production frontier from t to t + 1. Effch > 1 indicates an increase in relative efficiency change, while Effch < 1 indicates a decrease in relative efficiency change. Pech > 1 and Sech > 1 are positively correlated with Tfpch [33].

### 3.2. Sample

We select the sample (85 °C company) from listed companies of the bakery industry on Taiwan Stock Exchange (TSE) market. In Greater China, 85 °C represents one of the leading brands in the coffee–baking compound chain industry. 85 °C company is a multi-national enterprise with numerous stores in China, Hong Kong, Australia, Taiwan, and the United States. Until end of 2017, there were a total of 1074 85 °C chain stores around the world, such as Taiwan had 430 stores with 31 self-owned stores and 399 franchise stores, and China had 580 stores, which are all self-owned, whereas the United States had 40 self-owned stores; the Australia and Hong Kong had 15 and 9 stores respectively. The pre-tax net profit of Taiwan region accounted for 24.3% of the company's total profit in 2017 (85 °C internal information). After excluding the 9 self-owned stores without complete information from 2011 to 2016, the data were collected, taking into account the database of 22 self-owned stores (the proportion about 22/31 = 71%, in Taiwan total self-owned stores) from the 85 °C company database during 2011 to 2016 periods as samples. Similar services to customers are provided by these self-owned stores and use similar inputs.

### 3.3. Input–Output Variables

Based on data from the Taiwan Stock Exchange market and 85 °C company, several input/output variables, which consisted of the data for this study, had related to the operational characteristics of bakery (coffee) chain stores. The performance model of this study includes three outputs and three inputs. The output variables mainly consist of the following parts: Total revenue of bread (y1), total revenue of beverage (y2), and total revenue of pastry (y3) (includes incomes besides other two items such as souvenir sales, cake sales, snacking sales, and others). The input variables mainly consist of the following parts: Total staff salaries (x1) (includes all related pays of full-time employee such as salaries, labor premiums, pensions, healthcare expenses, and year-end bonuses), total dispatch employee salaries (x2) (includes all related pays of part-time employee such as salaries, labor premiums, pensions, healthcare expenses, meals, and year-end bonuses), and other expenses (x3), (includes rent cost and others). All outputs and inputs are measured in New Taiwan dollars (NT$).

### 3.4. DMU Quantity Rationality Test

In the use of DEA method, the number of decision-making units (DMU), the number of the inputs ($m$) and the number of outputs ($s$) are required to satisfy the formula, "DMU $\geq max\{m \times s,\ 3(m + s)\}$", which was proposed by Cooper, Seiford, and Tone [34]. In this article, the number of input indicators was 3, the number of output indicators was 3, and the empirical DMU sample is 22. Because" $22 \geq max\{3 \times 3,\ 3(3 + 3)\}$" indicates that the index has passed the rationality test.

## 3.5. Correlation Analysis of Input and Output Variables

The inputs (xi) and outputs (yi) correlation matrix are displayed in Table 1. Correlations with 5% significance show that most of the correlation coefficients between input and output are significant positive, and the isotonic property in this study are not been violated [35]. The result exhibits that these variables are appropriate for DEA modeling.

**Table 1.** Correlation analysis among variables.

|  | y1 | y2 | y3 | x1 | x2 | x3 |
|---|---|---|---|---|---|---|
| y1, total bread revenue |  | 0.099 | 0.495 ** | 0.512 ** | 0.669 ** | 0.782 ** |
| y2, total beverage revenue | 0.108 |  | 0.607 ** | 0.285 ** | 0.467 ** | 0.397 ** |
| y3, total others revenue | 0.456 ** | 0.531 ** |  | 0.412 ** | 0.700 ** | 0.741 ** |
| x1, total staff salary | 0.539 ** | 0.344 ** | 0.431 ** |  | 0.332 ** | 0.420 ** |
| x2, total dispatch worker salary | 0.664 ** | 0.376 ** | 0.604 ** | 0.334 ** |  | 0.691 ** |
| x3, total other expenses | 0.702 ** | 0.323 ** | 0.655 ** | 0.444 ** | 0.643 ** |  |

Note. (1) Upper-triangle number denotes the Person correlation coefficient; (2) lower-triangle number denotes the Spearman correlation coefficient; (3) ** $p$-value < 0.01.

## 3.6. Descriptive Statistics

Table 2 describes basic information of the sample per year for the six years in this study. The data shows that overall during the six-year period, there was growth in the mean for total revenue of beverage and pastry, however, the mean total revenue of bread was declined. In general, the mean for total revenue of beverage and pastry grew by 29% and 8% respectively between 2011 and 2016. The mean for total revenue of bread was, however, low at 11%. Overall, between the 2011 and 2016, the mean for total staff salaries and total dispatch employee salaries declined by 9% and 12%, respectively; mean for other expenses grew by 18%. The large-sized standard deviations of outputs may be caused by the significantly different sizes of self-owned stores.

**Table 2.** Means and standard deviations for the data of 85 °C (New Taiwan dollars, NT $).

| Year | 2011 | | 2012 | |
|---|---|---|---|---|
| **Variables** | **Mean** | **S.D.** | **Mean** | **S.D.** |
| Bread (y1) | 14,606,233.274 | 9,820,120.500 | 1,4107,033.410 | 9,132,508.323 |
| Beverage (y2) | 8,179,018.636 | 2,847,540.232 | 8,057,161.909 | 2,995,456.515 |
| Pastry (y3) | 7,038,586.274 | 2,293,196.144 | 6,621,458.592 | 2,157,413.557 |
| Staff salaries (x1) | 3,685,240.773 | 1,349,191.904 | 3,682,718.136 | 1,333,048.162 |
| Dispatch worker salaries (x2) | 2,234,723.182 | 790,718.199 | 2,223,295.227 | 882,616.308 |
| Other (x3) | 2,507,625.591 | 1,126,127.093 | 2,597,920.591 | 1,164,520.956 |
| **Year** | **2013** | | **2014** | |
| **Variables** | **Mean** | **S.D.** | **Mean** | **S.D.** |
| Bread (y1) | 1,784,660.637 | 9,139,500.227 | 12,785,729.951 | 8,963,175.280 |
| Beverage (y2) | 8,076,408.636 | 3,132,715.063 | 8,589,031.455 | 3,433,901.873 |
| Pastry (y3) | 6,682,305.490 | 2,168,255.529 | 6,799,731.095 | 2,241,104.390 |
| Staff salaries (x1) | 3,940,081.727 | 1,523,407.313 | 3,832,312.409 | 1,362,247.307 |
| Dispatch worker salaries (x2) | 2,245,105.591 | 903,977.250 | 2,255,602.410 | 898,911.332 |
| Other (x3) | 2,721,947.717 | 1,190,284.168 | 2,802,145.227 | 1,197,389.765 |

**Table 2.** *Cont.*

| Year | 2015 | | 2016 | |
|---|---|---|---|---|
| Variables | Mean | S.D. | Mean | S.D. |
| Bread (y1) | 12,209,959.682 | 8,295,798.244 | 13,012,269.182 | 7,778,714.678 |
| Beverage (y2) | 9,694,391.636 | 3,901,860.644 | 10,554,344.955 | 4,086,325.354 |
| Pastry (y3) | 7,301,201.455 | 2,441,502.128 | 7,600,426.546 | 2,816,755.747 |
| Staff salaries (x1) | 3,340,198.409 | 968,107.631 | 3,340,906.136 | 968,973.330 |
| Dispatch worker salaries (x2) | 2,183,596.364 | 906,609.124 | 1,966,470.136 | 740,156.899 |
| Other (x3) | 2,988,681.364 | 1,275,813.603 | 2,960,104.182 | 1,281,971.654 |

## 4. Empirical Results and Discussion

### 4.1. Overall Analysis

The relevant productivity change calculations have been carried out using the "DEA-solver" software package. Table 3 summarizes the Malmquist productivity index and its decomposition results. Five indicators of 85 °C performance are calculated annually. These changes include change in relative efficiency (Effch), technical progress (Techch), change in pure technical efficiency (Pech), change in scale efficiency (Sech), and change in total factor productivity (Tfpch).

**Table 3.** Average total factor productivity change and its decomposition of 85 °C, 2011–2016.

| Period | Effch | Techch | Pech | Sech | Tfpch |
|---|---|---|---|---|---|
| 2011–2012 | 1.009 | 0.954 | 0.994 | 1.015 | 0.963 |
| 2012–2013 | 1.000 | 0.991 | 0.996 | 1.004 | 0.991 |
| 2013–2014 | 0.995 | 1.008 | 0.994 | 1.001 | 1.002 |
| 2014–2015 | 0.992 | 1.581 | 1.001 | 0.991 | 1.568 |
| 2015–2016 | 1.011 | 0.738 | 0.990 | 1.021 | 0.746 |
| Geometric mean | 1.001 | 1.021 | 0.995 | 1.006 | 1.023 |

Table 3 shows the higher Malmquist productivity index (= 1.023), indicates an average annual productivity growth rate of 2.3% in the 85 °C between 2011 and 2016. In addition, the relative efficiency was above 1 (Effch > 1) and technical progress was above 1 (Techch > 1), revealing that, for the 2011–2016 period, the management and organization of these self-owned stores had been improved as well as new technologies or innovations had been invested. Moreover, the increase in productivity was mainly due to technical progress, which increased by 2.1% per year (Techch = 1.021). This means that changes in total factor productivity are mainly the result of technical progress, rather than changes in relative efficiency. Thus, 85 °C experienced a high degree of technical progress during the study period, but achieved only modest productivity growth. Furthermore, the change of relative efficiency has a positive impact on the change of total factor productivity, which is mainly shown as the annual change rate of pure technical efficiency decreases by 0.5%, and the annual change rate of scale efficiency increases by 0.6%.

As displayed in Figure 1, 2011–2012, 2012–2013, 2013–2014, 2014–2015, 2015–2016 periods of relative efficiency change index were 1.009, 1.000, 0.995, 0.992, and 1.011. Relative efficiency change first decreased and finally increased, showing an uptrend after a decline. The technical progress first increased and finally decreased, displaying a decreasing pattern after increasing. Pure technical efficiency first goes up, then down, then up, then down. The change trend of scale efficiency was the same as that of technical progress. The change trend of total factor productivity was consistent with that of technical progress.

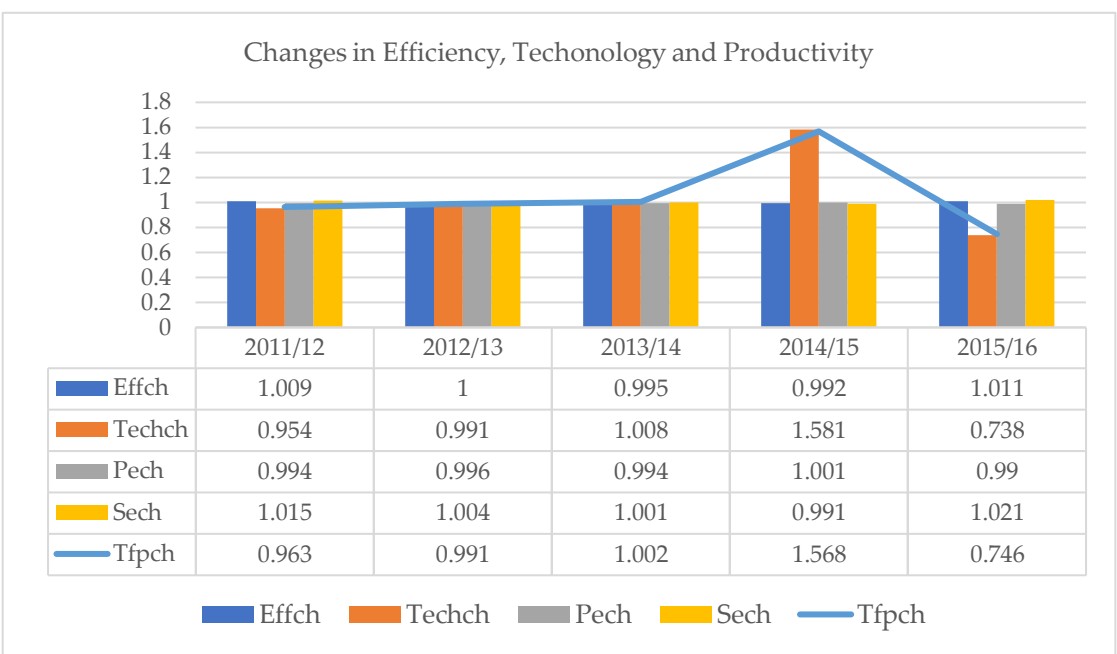

**Figure 1.** Trend of productivity change and its component in 85 °C from 2011 to 2016.

*4.2. Analysis of Different Districts*

Since the latitude of Taiwan is on the Tropic of Cancer, and there are tropical and subtropical climates, it is necessary to further understand whether the total factor productivity changes of each self-owned store will be different depending on the location of the self-owned store. Therefore, the regions where the self-owned stores are located are divided into three districts of north, central, and south based on the geographical division of the Tropic of Cancer, and the change of the productivity of self-owned stores in the three districts is further discussed in this paper.

Table 4 shows that, based on the output-oriented DEA-Malmquist productivity index, an estimate of the change in total factor productivity (Tfpch) and its components, i.e., relative efficiency (Effch) (catch-up component) and technical progress (Techch) (innovation or frontier-shift component) during the period 2011–2016. The study consisted of analyzing which part of the total productivity factor change can be attributed to relative efficiency, and which part to technical progress, causing a shift in the production frontier. Relative efficiency requires the decision-making unit (DMU) to allocate resources without wasting. These requirements mean, for example, improving self-owned store management and organization better investment plans, and improving technical expertise. Technical progress reveals a change in technology as a consequence of innovation and of new technologies being adopted by best practices self-owned stores.

On average, DMU 20 had the highest Tfpch growth rate at 10.6% and DMU 13 had the lowest Tfpch growth rate at 7.7%. Overall, the average change of TFP over this period was 1.023, which implies a 2.3% increase in Tfpch at 85 °C. In the 2011–2016 period, 16 self-owned stores out of 22 had positive efficiency changes (Tfpch > 1), mostly because of positive technical progress, indicating an improvement in technology and innovation in these self-owned stores (Table 4). In addition, Table 5 shows the changes experienced by 22 self-owned stores in all efficiency categories over the past 6 years by direction and number. It may be said that 6 self-owned stores in relative efficiency, 16 self-owned stores in technical progress, 2 self-owned stores in pure technical efficiency change, 9 self-owned stores in scale efficiency change, and 16 self-owned stores in total factor productivity change achieved improvements for the study period. Comparing numerically, 73% of the self-owned stores showed significant improvements in the level of technical progress and changes in total factor productivity, which is similar result of Tatli and Bayrak [12].

**Table 4.** The summary of Malmquist productivity index of 85 °C, 2011–2016.

| District | DMU | Effch | Techch | Pech | Sech | Tfpch |
|----------|-----|-------|--------|------|------|-------|
| North | 1 | 0.996 | 1.005 | 1 | 0.996 | 1.001 |
| Central | 2 | 1.028 | 1.033 | 1.007 | 1.02 | 1.061 |
| Central | 3 | 1.02 | 1.046 | 1 | 1.02 | 1.066 |
| North | 4 | 1 | 1.011 | 1 | 1 | 1.011 |
| North | 5 | 1 | 0.987 | 1 | 1 | 0.987 |
| North | 6 | 0.982 | 0.999 | 0.977 | 1.006 | 0.982 |
| South | 7 | 1 | 1.085 | 1 | 1 | 1.085 |
| Central | 8 | 1.034 | 1.041 | 1 | 1.034 | 1.076 |
| Central | 9 | 0.983 | 1.069 | 0.968 | 1.016 | 1.051 |
| Central | 10 | 1 | 1.079 | 1 | 1 | 1.079 |
| Central | 11 | 1 | 1.013 | 1 | 1 | 1.013 |
| North | 12 | 0.996 | 1 | 1 | 0.996 | 0.996 |
| North | 13 | 0.968 | 0.953 | 0.972 | 0.996 | 0.923 |
| North | 14 | 0.981 | 1.003 | 0.978 | 1.003 | 0.983 |
| North | 15 | 1 | 0.992 | 1 | 1 | 0.992 |
| North | 16 | 1 | 1.003 | 1 | 1 | 1.003 |
| South | 17 | 1.006 | 1.005 | 0.982 | 1.025 | 1.012 |
| North | 18 | 1 | 1.048 | 1 | 1 | 1.048 |
| North | 19 | 1 | 1.023 | 1 | 1 | 1.023 |
| South | 20 | 1.06 | 1.043 | 1.055 | 1.005 | 1.106 |
| South | 21 | 0.956 | 1.051 | 0.956 | 0.999 | 1.005 |
| North | 22 | 1.023 | 0.994 | 0.997 | 1.026 | 1.017 |
| Geometric mean | | 1.001 | 1.021 | 0.995 | 1.006 | 1.023 |

**Table 5.** Variation of Efficiencies of 22 self-owned stores, 2011–2016.

| | Effch | Techch | Pech | Sech | Tfpch |
|----------|-------|--------|------|------|-------|
| Increased (>1) | 6 (27%) | 16 (73%) | 2 (9%) | 9(41%) | 16 (73%) |
| Stable (=1) | 9(41%) | 1 (4%) | 13 (59%) | 9(41%) | - |
| Decreased (<1) | 7 (32%) | 5 (23%) | 7 (32%) | 4(18%) | 6(27%) |

On the one hand, Table 6 indicates the results of relative efficiency change, technical progress, and total factor productivity change in different operation districts. The results of the non-parametric Kruskal–Wallis test (K-W test) and the mean productivity change levels for the three operation districts of 85 °C self-owned stores during the 2011–2016 period are shown in Table 6. It is clear that in Table 6 that the north-district compared with central-district and south-district, is less relative efficiency change level (Effch = 0.9955), technical progress (Techch = 1.0015) and total factor productivity change level (Tfpch = 0.9972).

In addition, there is a difference of a significance level of 5% between the total factor productivity change level and technical progress (Tfpch and Techch) for the three operation districts of 85 °C self-owned stores, by using KW-test in the Table 6. Moreover, it reveals that the north-district self-owned stores have the worst technical progress, and total factor productivity change level during 2011–2016 period, after performing the Dunn post-hoc test. This may be due to the fact that the self-owned stores in the north district are located in subtropical climate, and have more competition from other bakery stores.

**Table 6.** Average change in TFP and its different components, and the Kruskal–Wallis test.

| Districts | Effch | | Techch | | Tfpch | | Sample Number |
|---|---|---|---|---|---|---|---|
| | Average | R-Mean | Average | R-Mean | Average | R-Mean | |
| North | 0.9955 | 9.42 | 1.0015 | 7.29 | 0.9972 | 7.33 | 12 |
| Central | 1.0108 | 14.7 | 1.0468 | 16.50 | 1.0577 | 17 | 6 |
| South | 1.0055 | 13 | 1.0460 | 16.63 | 1.0520 | 15.75 | 4 |
| Kruskal-Wallis test (Z) | 3.086 | | 11.11 | | 10.958 | | |
| *Prob.* > Z | 0.214 | | 0.004 | | 0.004 | | |
| Dunn post-hoc test (Q) | — | | North < Central, South | | North < Central, South | | |

Note: R-mean denotes the mean of rank.

### 4.3. Management Decision Matrix

In order to further propose the management significance of relative efficiency change and total factor productivity change, the study constructed a management decision matrix based on relative efficiency change and total factor productivity change similar to BCG matrix, so as to facilitate the decision-making of managers. The relevant practices are explained as follows.

Based on Table 4, the article identified four combinations of Tfpch and Effch, dividing 85 °C self-owned stores among the corresponding quadrants as Figure 2. The change of total factor productivity (Tfpch) of each self-owned store from 2011 to 2016 is used to reflect the sustainable development potential of each self-owned store, and the relative efficiency change (Effch) of 2015–2016 can show the spot competitiveness. The Tfpch of each self-owned store from 2011 to 2016 is placed on the vertical axis, and the relative efficiency change from 2015 to 2016 is taken as the horizontal axis, which is divided into four quadrants (Figure 2). The choices about the cut-off points are as follows: The cut-off point of Tfpch is 1, the value greater than 1 indicates productivity (performance) progress, and the value less than 1 indicates productivity (performance) deterioration. The cut-off point of the horizontal axis is the mean value of the relative efficiency change of all self-owned stores. The quadrant in which both the x-coordinate and y-coordinate are greater than the value of the cut-off point is defined as the first quadrant. In clockwise order, the specific meaning of each quadrant is as follows:

1. The seven self-owned stores (DMU 1, 7, 8, 17, 19, 20, and 21) located in the first quadrant are excellent in terms of long-term potential competitiveness and spot competitiveness. The self-owned stores located in this region are competitive ones with better productivity (performance). They not only have good long-term development potential, but also have better operation efficiency (relative efficiency) than other self-owned stores in the past two years.
2. A self-owned store (DMU 6) located in the second quadrant has a good operating performance at present, but the long-term productivity change is not satisfactory, indicating that the self-owned store located in this quadrant is competitive at present, while the long-term productivity change is bottleneck.
3. The five self-owned stores (DMU 5, 12, 13, 14, and 15) located in the third quadrant are at a competitive disadvantage in the total factor productivity and current operating performance. They not only need to improve in terms of operating performance, but also in terms of productivity growth.
4. The nine self-owned stores (DMU 2, 3, 4, 9, 10, 11, 16, 18, and 20) located in the fourth quadrant have a certain level of total factor productivity change, but there are areas where the current operation performance needs to be improved. From the perspective of operating performance, the self-owned stores located in this region can further adjust their operating efficiency, and over time, they will be further promoted to be competitive and have technical progress advantages. Therefore, self-owned stores located in this quadrant have competitive advantages.
5. To sum up, this paper suggests that 85 °C enterprises should take the self-owned stores in the first quadrant as the learning benchmark for the self-owned stores in other quadrants.

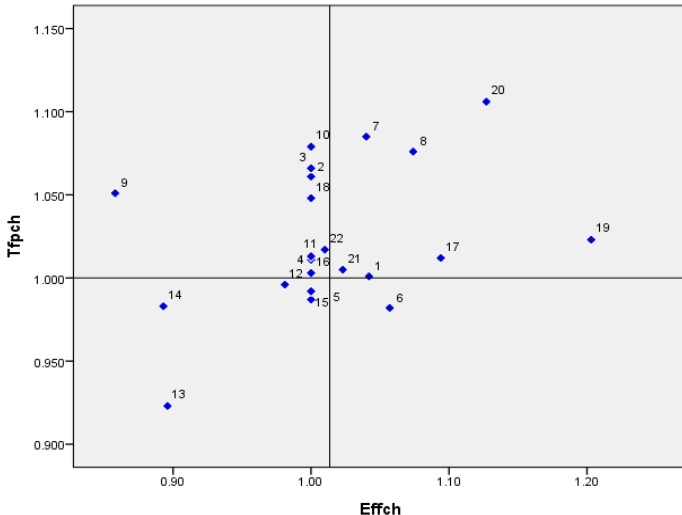

**Figure 2.** Management decision matrix of 85 °C during 2011–2016 periods.

Brief summary, the significance and differences between this paper and the existing research are: (1) To the authors' knowledge, most of the studies so far have focused on productivity changes in food manufacturing in different countries or regions, and as using DEA to measure bakery industry productivity change have not been found. This study first time analyze by applying DEA-Malmquist model to analyze the productivity growth in baking industry; (2) this study not only expands the previous static efficiency researches [9], but also emphasizes the dynamics, making the research results more representative and generalized; (3) this paper carries out dynamic analysis for the efficiency by year with the uses the Malmquist index model to analyze the dynamic change of efficiency of 85 °C company, and discusses the patterns of the efficiency of 85 °C company, and further broaden the research perspective; (4) on the basis of the research results of the productivity change of 85 °C company, it provides some other deep viewpoints for management of bakery companies so as to promote the sustainable development of this industry.

## 5. Conclusions, Limitations, and Future Research

This paper used the DEA-Malmquist model with three inputs and three outputs to investigate the productivity growth of 85 °C company from 2011 to 2016. The empirical results demonstrate as a result of a positive change in technical progress and a positive change in relative efficiency, the overall mean for total factor productivity change (Tfpch) was positive at 1.023. It implies that the productivity growth in this industry is contributed by positive change in technical progress as well as the gain in relative efficiency. The growth of Tfpch in 2014–2015 was the highest of 56.8%, while the growth of Tfpch in 2015–2016 was the lowest of −25.4%. The change of total factor productivity is mainly the result of technical progress rather than relative efficiency change. It is necessary to encourage technical innovation to ensure faster technical progress in 85 °C company.

In addition, it reveals that the north-district self-owned stores, compared with central-district and south-district, has the worst technical progress level, and total factor productivity change level during 2011–2016 period by performing KW-test and the Dunn post-hoc test. Moreover, among the 22 self-owned stores, 15 of them show positive change in relative efficiency, and 12 of them also show positive change in technical progress. In other words, the 12 self-owned stores had made positive technical progress and relative efficiency changes. These findings could benefit self-owned store managers to improve their performance, which could be used as a benchmark for the best-performing self-owned stores.

To sum up, a practical framework to support academics and practitioners has been introduced by this paper. Academically, the contribution of this study lies in applying DEA-Malmquist model to the

productivity change analysis of baking industry for the first time. There have been no case studies using DEA to measure bakery industry productivity change before. Therefore, a pioneer reference for similar research in the future has been provided by this study. From a practical implications point of view, productivity change analysis offers some advantages from a management perspective. Managers are able to identify strategic importance of self-owned stores by adopting the Malmquist productivity index. In a highly competitive environment, these self-owned stores where relative efficiency change occurred in tandem with technical progress appear to be the ones with highest abilities and inherent capability to thrive instead of just survive.

The study had several limitations and future directions. First of all, this paper focuses on case studies and takes 85 °C self-owned stores in Taiwan as the research object to analyze the changes of its productivity. The generalization of the result is finite. In the future, in order to provide more insight into the performance of this industry and the differences between countries or regions, future studies will integrate data from 85 °C branches in different countries (China, USA, and Australia) or regions (Hong Kong), focusing on the development of indicators of productivity changes to assess the meta-frontier of companies, to generalize the empirical results of this article. In addition, to gain additional insights, the results of this study could be compared with Taiwan Stock Exchange (TSE) other bakery companies or other country bakery companies. Secondly, another restriction of this study was the types and numbers of input and output variables, future research may refine the input-output structures according to the needs of customers in different regions.

**Author Contributions:** C.-W.C. and K.-S.W. planned the study and wrote the article. C.-W.C., K.-S.W., and B.-G.C. explored the data. C.-W.C., K.-S.W., and B.-G.C. collected and treated the data. All the authors have read and approved the final manuscript.

**Funding:** There was no external funding for the study.

**Conflicts of Interest:** The author states that there is no conflict of interest.

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
