# Peer review of "Productivity Change and Decomposition in Taiwan Bakery Enterprise―Evidence from 85 °C Company"

_sustainability, doi:10.3390/su11247077_

Round 1

Reviewer 1 Report

All the inputs and outputs are expressed in monetary terms, hence I see no point in using DEA for this exercise. There is no literature provided to support the choice of the inputs and outputs.

Author Response

please see in the attachment.

Reviewer 2 Report

Authors accepted all the changes I required. 

Now this paper can be published.

Author Response

Dear Reviewer:

Thank you for your affirmation.

Reviewer 3 Report

I thank the authors the effort made for the improvement in the paper.

I only recommend a correction of minor mistakes:

Line 137, shown instead of showd; line 204 with an unnecessary full stop; line 206 taken instead of taking. Titles of the tables should be estandardized. Figure 1 must be changed, the information is not clear.

Author Response

please see in the attachment.

Round 2

Reviewer 1 Report

I do not recommend publishing this paper in Sustainability. The data used are not related to sustainability. There is no contribution to the existing literature (except that bakery in Korea is analyzed for the first time).

This manuscript is a resubmission of an earlier submission. The following is a list of the peer review reports and author responses from that submission.

Round 1

Reviewer 1 Report

The parper should be improved. My suggestions are the following:

1- Regarding the introduction section, the study should be linked to sustainability topic. The relationship of the aim of the study with this topic is week. 

2- The authors state in introduction section: 

...its productivity growth can significantly affect the profitability of Taiwanese companies and the welfare of consumer.

Please, expand this statement, try to find the connection with sustainable development.

3- The introduction section lacks a clear contribution. I suggest explaining in a better way the usefulness of this study.

4- With regards to section 3, Materials and Methods, the subsection sample should be enriched. More details about 85º C should be provided. For expample,the percentage or proportion that the 22 self-owned stores account for the population (total stores in the country). 

5- All tables should be improved according to the presentation. For example, table 2 lacks tittle and the information should be shown in a simpler way. I recommend other type of table.

6- The spelling should be reviewed.

Reviewer 2 Report

The paper has a fair potential of interest for the readers of sustainability. The contents of this paper are quite in line with the editorial objectives of Sustainability, despite it does not contain specific topics relating to the sustainability of productions.

I believe that paper could be published but authors should ammeliorate any parts of present paper, since it presents some shortcomings concerning results and discussion.

Introduction

It is quite organized and well presented

Methodology

Pag. 4 Line 175: I suggest to change “this article takes” ….. into “data were collected took into account the database of 22 self-owned……

Literature review:

It is quite complete. I only suggest to entitle this paragraph as “Literature review on Malmquist total factor productivity index”

Discussion:

Authors should compare their results with previous studies and outcomes published by other authors. I believe that authors should to emphasize the significance of obtained results and discuss them in depth and eventually to open new questions to be solved.

No effective reference to the existing literature was made. It is necessary a deepen discussion bringing more concepts and citing previous contribution on the same field.

Conclusion:

Authors should discuss more in depth general implications for producers and academicians, trying to explain how the empirical results could be reasonably extended to other national contexts.

Reviewer 3 Report

The methods applied in the paper are all conventional ones. The empirical analysis is not convincing. It is unclear why this company should be covered in the analysis. Thus, the research falls below the standards of an international SSCI journal.

The concept of sustainability is not addressed in the analysis. Thus, the paper does not fit the scope of sustainability.

The technology is defined in a strange manner as the variable are all measured in monetary terms (mostly cash). Thus, simple ratio analysis is enough.

The results show no TFP change. Thus, the message of the paper is unclear.

The quality of presentation is not acceptable (e.g. English language, tables and figures are not informative).